# Smart Artificial Markers for Accurate Visual Mapping and Localization

**DOI:** 10.3390/s21020625

**Published:** 2021-01-18

**Authors:** Luis E. Ortiz-Fernandez, Elizabeth V. Cabrera-Avila, Bruno M. F. da Silva, Luiz M. G. Gonçalves

**Affiliations:** Natalnet Associate Laboratories, Campus Universitário, Federal University of Rio Grande do Norte, Natal RN 59078-970, Brazil; lortiz@dca.ufrn.br (L.E.O.-F.); vcabrera@dca.ufrn.br (E.V.C.-A.); bruno.silva@ect.ufrn.br (B.M.F.d.S.)

**Keywords:** smart markers, visual mapping, visual localization

## Abstract

Artificial marker mapping is a useful tool for fast camera localization estimation with a certain degree of accuracy in large indoor and outdoor environments. Nonetheless, the level of accuracy can still be enhanced to allow the creation of applications such as the new Visual Odometry and SLAM datasets, low-cost systems for robot detection and tracking, and pose estimation. In this work, we propose to improve the accuracy of map construction using artificial markers (mapping method) and camera localization within this map (localization method) by introducing a new type of artificial marker that we call the smart marker. A smart marker consists of a square fiducial planar marker and a pose measurement system (PMS) unit. With a set of smart markers distributed throughout the environment, the proposed mapping method estimates the markers’ poses from a set of calibrated images and orientation/distance measurements gathered from the PMS unit. After this, the proposed localization method can localize a monocular camera with the correct scale, directly benefiting from the improved accuracy of the mapping method. We conducted several experiments to evaluate the accuracy of the proposed methods. The results show that our approach decreases the Relative Positioning Error (RPE) by 85% in the mapping stage and Absolute Trajectory Error (ATE) by 50% for the camera localization stage in comparison with the state-of-the-art methods present in the literature.

## 1. Introduction

A recurring problem in computer science is the estimation of the 3D pose (position and orientation) of an optical sensor that is capturing images of an environment. Different solutions for the 3D camera pose estimation problem, as it is known in the literature, are actively sought by the robotics [1,2], computer vision [3], virtual (augmented) reality [4], and photogrammetry [5,6,7] communities, to name a few. The proposed systems and algorithms have different performance requirements, according to the problem domain. For example, algorithms proposed for augmented reality aim to estimate camera pose while minimizing drift (accumulated pose error) and jitter (pose oscillation) at the same rate that images are acquired (e.g., 30 frames per second). This improves the sense of immersion for the user of that system [4]. A considerable portion of studies in computer vision and automated photogrammetry [3,8] have as their main goal improving the accuracy of the 3D models built from images. In photogrammetry [5,6,7], accuracy is a mandatory requirement. However, the methods involved have a high associated cost, and most of the models cannot be directly applied in real-time applications such as robotics and virtual reality.

In robotics, the main requirement is that the robot should be able to compute its 3D pose relative to a global reference frame in real-time and with enough accuracy to allow its operation in a fully autonomous manner. This problem is so relevant that it has a denomination, being known by SLAM (simultaneous localization and mapping) [9] in the robotics community. To solve SLAM, proposed algorithms build a map of the environment at the same time that the robot is localized within this map. For this, feature extraction [10,11] plays an important role, since a large portion of camera pose estimation methods are based on the use of natural features known as keypoints [1,12].

Nonetheless, in robotics applications that require more accurate pose estimation without abdicating real-time performance, the exclusive use of a pipeline based on natural feature extraction and matching is not sufficient. Keypoint extraction and matching algorithms may be often confused by repetitive visual patterns or the total absence of visual features in images. Additionally, the invariability to rotation and scale changes inherent to keypoint extraction algorithms, such as SIFT (scale invariant feature transform) [13] and SURF (sped up robust features) [14], is only partial and cannot be guaranteed for every image.

Research has then been proposed aiming for more accurate camera pose estimation with the integration of other techniques. Visual inertial SLAM [15] algorithms fuse visual data from images with data gathered from other sensors, such as inertial measurement units (IMUs). Another example is the integration of SLAM with the automatic detection of fiducial markers [16,17] in the form of black and white squares, widely known as artificial markers in the augmented reality community [18,19]. Notice that these are not marks of the same kind present in photogrammetric images, which are named fiducial marks [20]; the inspiration came from there. As employed by the latter class of methods, these artificial markers aid image-based SLAM algorithms by providing easy and unambiguous detection of reference objects used in optimization routines. This in turn results in a more accurate camera poses than those computed solely from image data without artificial markers.

Nevertheless, the resulting upper bounds for the errors of camera poses obtained with artificial markers remain a problem. Some applications, such as the use of artificial markers as ground truth for a SLAM dataset [21], still lack the required accuracy. In the PennCOSYVIO application [21] the authors report a ground truth built solely with artificial markers with a Relative Positioning Error (RPE) of approximately 10 cm in the translation components.

In light of this discussion, this work proposes a camera pose estimation system based on a novel type of artificial marker called the smart marker. A smart marker consists of a square planar fiducial marker and a pose measurement system (PMS), which is built with a distance sensor and an IMU. These markers can easily be built by extending existing square fiducial marker systems (e.g., ArUco [18,19]) with the required electronic components. The system works using visual information gathered from consumer-grade perspective cameras and PMS data collected from a set of smart markers distributed throughout the environment of operation. It is thus expected that the 3D poses of cameras capturing images of smart markers may be computed with improved accuracy in comparison with the use of visual SLAM solutions or purely visual marker-based pose estimation methods. We opted to rely on artificial markers instead of natural keypoints (or tie points, as known in photogrammetry [7]) to avoid three sources of significant error: (1) errors originated from erroneous tracking of keypoints, (2) errors due to pure rotational camera movements and (3) errors present in scenes without texture. This latter type of error occurs in some indoor environments and is known to affect keypoint detection, and consequently, camera pose estimation.

The operation of the smart markers for 3D camera pose estimation is enabled through the use of the graph optimization framework [22,23]. This framework allowed us to design a sensor fusion algorithm to merge the data from all involved sensors and build an optimized graph of poses consisting of the global 3D poses of all smart markers present in the environment. More precisely, a pose graph was built having as vertices the 3D poses of each marker and as edges relative measurements (relative poses, orientations, or translations) between a given pair of markers. Since there will be redundant information (i.e., more than one path between two assorted nodes of the graph), the sum of the errors on all edges can be minimized to yield a graph containing the optimal poses of the smart markers. Although similar to bundle adjustment [24,25], the optimization algorithm minimizes the sum of 3D alignment error present on all edges instead of the reprojection error of keypoints on the images.

Accordingly, the proposed method can estimate the camera poses with an algorithm divided into two stages. Firstly, the pose graph of all smart markers and cameras is constructed from the data collected from images and PMS units and optimized to minimize the total error within the graph. We call this first stage the mapping stage, which computes a map in the form of a network of 3D rigid transformations, each of them attached to a smart marker (i.e., the graph after optimization). In the second and online stage, images of the smart markers are captured and another optimization procedure is executed to compute the 3D pose of the camera in relation to a known set of (previously optimized) smart markers. The minimization procedure operates in a pose graph similar to the first stage, except that only the poses of detected smart markers in the image of the camera to be estimated are considered, forming a subgraph of the complete network of transformations. Since each detected marker gives a hypothesis for the camera pose, the localization method computes the camera pose that minimizes the errors relative to all measurements in the least-squares sense. This second stage is called the camera localization stage. In essence, the camera localization stage assumes a camera with known intrinsic parameters [26] and computes the camera pose based on known 3D structure. This has the same goal as the perspective-n-point problem [27] (as known in the computer vision community) or the space resection problem [7] (as known in the photogrammetry community), although in our method the camera pose is obtained considering the 3D pose of the smart markers instead of considering 3D points.

Therefore, two novel contributions are present in this paper: (a) the smart marker device and (b) the two-stage algorithm for multimodal sensor fusion and camera pose estimation.

To validate our proposal, the mapping stage is compared to the Marker Mapper [28] algorithm and to ground truth gathered by hand with more precise sensors. Similarly, the camera localization stage is compared with the following SLAM methods: LibViso2 [29], which uses natural keypoints; UcoSLAM [17] configured to use natural keypoints and artificial markers; and the commercial 6-DoF pose estimation method available for the Stereolabs ZED camera [30]. The results obtained in the experiments provide enough evidence of the usefulness of the method. As will be demonstrated in the following sections, our method decreases the RPE by ≈85% in the mapping stage and the absolute trajectory error (ATE) in ≈50% in the camera localization stage. Additionally, the results support the claim that it is possible to develop low-cost applications for robot pose estimation, including but not limited to ground truth generation to benchmark visual SLAM algorithms.

The rest of the article is structured as follows. Section 2 lists the related work, including the state of the art, and Section 3 formulates the problem. In Section 4 our approach is introduced, and Section 5 discusses the experiments performed for validating it. Finally, we present a discussion of the results in Section 6; the limitations of our method are discussed in Section 7. Finally, the conclusions are in Section 8.

## 2. Background and Related Work

Methods based on fiducial markers are very popular for camera pose estimation with acceptable precision and speed. Additionally, several fiducial marker types have been proposed in the literature. In this section, we discuss the accuracy and the adopted term fiducial markers, and then the related work on fiducial markers themselves—the work using sensors’ and fiducial markers’ data fusion, and the work on mapping and localization using fiducial markers that are all closely related to our work.

### 2.1. A Note on the Accuracy of Photogrammetry

In photogrammetry [5,6,8,31] the main goal is to improve the accuracy of the 3D models built from images. Hence, accuracy is a mandatory requirement. This type of application relies on highly-controlled images acquired from specific cameras in general with a high-precision capture process—with a high associated cost, however. Some of the techniques can be of inspiration and even adapted for lower-cost equipment such as cell phones and cameras, and the community has done it since the appearance and first uses of digital photography [8,32]. Nonetheless, most of the models cannot be straightforwardly applied in real-time applications such as robotics and virtual reality. In these real-time applications, precision is desired, but it can be neglected to a certain degree for running in real-time, and in general online processing data using lower-cost equipment is permitted.

Cartographic or 3D maps, which are results from photogrammetry, should be as perfect and accurate as possible, as they will generally be used for high-precision control. They are in general produced following a series of steps after image acquisition [5,6], in a very well controlled, currently automated pipeline of production for accounting for the several issues regarding their precision and images resolution. Oppositely, as said above, in this work we deal with online, real-time applications, so a different approach than the one used in photogrammetry (and cartography) should be applied here as we do not have time to post-process data. That said, one can notice that photogrammetry techniques, besides being well settled in the literature cannot be straightforwardly applied in the robotic vision problem, which is the topic here.

Additionally, the concept of fiducial marks and ground control points (GCPs) in photogrammetry should be introduced to disambiguate from the one commonly used in augmented reality [16,17], which is called fiducial markers. In photogrammetry, fiducial marks are small crosses or small V-shaped indents located precisely on each of the four corners or exactly midway along the four sides of a scanned-film photograph. After one identifies the fiducial marks in a scanned image, they can be used with the fiducial marks entered from the camera calibration report to establish an image coordinate system. In general, the coordinates of the fiducial marks are a compulsory parameter for scanned-film photographs. Images taken with digital cameras and cell phones do not contain fiducial marks [20]. GCPs are elements present (or introduced) in the scene, with known world coordinates, to provide the exterior orientation [5,6,7]. These elements can be natural (house corners, triangulated points) or artificial markers that are put into the scene with coordinates measured. In this work and in augmented reality the term fiducial markers refer to planar markers that are generally built in the form of black and white squares, also known as artificial markers [18,19].

It is also important to mention that modern photogrammetry [7,33] and the computer algorithms designed for its automation [8] make extensive use of non-visual data to meet the accuracy requirements. For example, the integrated sensor orientation [33] approach proposes the use of IMU and GPS in a single encapsulation for photogrammetry applications. Although similar to our approach of fusing non-visual data with images, a substantial difference lies in the fact that we employ IMU and distance sensors on artificial markers, not on the image acquisition device. We discarded using GPS on the smart markers because GPS signals are not available in indoor scenarios.

### 2.2. Camera Pose Estimation from Images

The estimation of camera pose from visual data is a well known and studied problem in computer science. This study is mainly driven by virtual (augmented) reality [4], computer vision [34,35], and robotics [1] applications. In fact, the problem has solutions that are not exclusive to optical sensors [36].

Real-time camera pose estimation methods based on natural features (e.g., salient keypoints such as SIFT [13] or SURF [14]) with known 3D positions or based on known 3D objects [4,34] have been proposed, and they are commonly referred to as camera tracking methods. In addition to the use of known 3D/2D point correspondences, these methods make use of RANSAC [27] to estimate mathematical models from projective geometry [25,26] by excluding possible outlier data.

The problem has also been investigated without assuming known 3D structure or camera geometry, in what is known as structure from motion (SfM) [3,35]. These systems output 3D models in the form of dense textured meshes or dense point clouds as their final results. For this, keypoint extraction and matching, along with RANSAC, are extensively used to obtain an initial estimate for the imaged 3D structure and also for the relative transformations between cameras. Similarly to photogrammetry, the main goal of SfM is obtaining accurate 3D models from images, and therefore, real-time performance is not achievable. To meet such strict accuracy requirements, the initial estimates for 3D structure and camera poses are optimized in bundle adjustment (BA) [24,25]. BA optimizes 3D points and camera poses (and optionally the camera intrinsic parameters [26]) by minimizing the sum of reprojection errors, obtained by the differences in image coordinates between the keypoint and its predicted image coordinates. These predictions are computed by projecting the 3D points into the image from which the keypoints were extracted using the initial camera pose and camera calibration as the projection parameters. A gradient-based search for optimal 3D structure and camera poses yield the (non-linear) least-squares solution for the problem. Since the number of optimized parameters in SfM applications is in general large, BA computation of the full problem is not feasible in real-time.

In the last two decades, SfM tools have been adapted to work with real-time performance. Needless to say, the required changes incur less accuracy than those obtained with SfM. Methods proceeding in this direction are known as visual odometry [37] and enable real-time 3D reconstruction by processing temporal sliding windows with 3D points and camera poses to be incrementally optimized.

The advent of more precise visual sensors made it possible for the design of algorithms focused on optimizing camera poses only without considering 3D points for the optimization [1]. With this, the optimization engines commonly built based on BA were exchanged by graph optimization engines [22,23], which employ the same numerical routines used by BA to minimize the accumulated error on a pose graph. In turn, this resulted in systems obtaining more accurate 3D reconstructions yet with real-time performance. Since the graph optimization framework is very general by construction, it is possible to formulate BA solutions within this framework with the appropriate tools (e.g., g2o [23]). Currently, visual odometry has matured to integrate visual SLAM algorithms, as proposed by the current state-of-the-art feature-based visual SLAM algorithms [38,39,40] or visual-inertial SLAM [15,41,42,43,44,45], which works by fusing visual information with IMU, LIDAR, ToF, GPS or mechanical odometry data.

### 2.3. Fiducial Markers

The Fourier tags [46] approach features a radial sinusoidal pattern, in which it stores PSK-encoded information. It uses the circular shape of the marker to calculate its pose. Other works use circular planar markers where the identification is encoded in concentric rings of dots or circular sectors [47,48]. CCTag [49] relies on the number of gradient magnitudes through an arc of the outermost ellipse and one of the inner-most ellipses to deal with occlusions due to motion blur and illumination changes. However, circular markers usually provide just one correspondence point (the center), making the detection of several of them necessary for camera pose estimation.

An alternative to these previous approaches is a square planar marker that has been used in the field of virtual (augmented) reality for many years. These markers each consist of an external black border and an internal binary code for identification. Their main advantage is that a single marker provides four correspondence points for pose estimation. ARToolKits [50] is one of the first that uses square fiducial markers. It employs markers with a custom pattern that is identified by template matching. The identification process has several limitations, such as high computational time, a high number of false positives, and considerable sensitivity to varying lighting conditions. BinARyID [51] introduces the generation of customizable marker codes for identification but does not consider the possibility of error detection and correction. ARToolkitPlus [52] allows the use of error detection and correction techniques for occlusions. However, one of its main drawbacks is that the dictionary of markers is fixed, and thus, optimal results in error detection and correction capabilities are not achieved.

Despite the significant advances achieved so far, fiducial markers have some limitations. If the marker is partially occluded, pose estimation cannot be done and the fixed size of the marker makes it impossible to be detected under a wide range of distances. Some authors have proposed alternatives to overcome the above issues. ArTag [53] handles the partial occlusion using an edge-based method. Edge pixels are thresholded and connected in segments, which are grouped into sets and used to create a mapping homography. Nevertheless, markers cannot be detected when more than one edge is occluded and the method is not considered suitable for real-time applications. ChromaTag [54] presents a fiducial marker that uses concentric inner red and green rings for robust color variation detection.

The most recent work of ArUco [18,19] introduced a method for fast and accurate detection of fiducial markers. The method uses image segmentation for increased speed and a multi-scale image to find the positions of the marker corners with sub-pixel accuracy.

### 2.4. Using Sensors and Fiducial Markers Data Fusion

Data fusions between fiducial markers and other sensors are also proposed in the literature. For example, there are monocular visual-inertial SLAM approaches that fuse 6-DoF inertial measurements and observations of AprilTags using Extended Kalman Filter (EKF) [55,56]. For these methods monocular and stereo cameras with embedded IMUs are used, respectively, with these sensors mounted on the top of mobile robots.

Lopez et al. [57] present a method to fuse inertial information with visual information of fiducial markers for state estimation in aerial robots. In the same form, a multi-sensor fusion for indoor localization based on the ArUco marker is presented by Xing and colleagues [58]. This method uses a camera and sensors (inertial and ultrasonic) mounted in a micro aerial vehicle (MAV).

### 2.5. Mapping and Localization Using Fiducial Markers

Methods based on natural keypoints suffer from some limitations; for example, they fail in the case of pure rotational movements and require a certain amount of texture that is not always available in indoor environments.

An alternative to tracking natural keypoints in images is detecting fiducial markers. Klopschitz and other authors [59,60,61] introduced methods for solving visual SLAM based on markers. However, they do not consider optimizing the estimated marker poses for the ambiguity problem. The ambiguity in planar marker pose estimation arises when more than one plausible pose solution for the marker pose is possible, which happens due to noisy coordinates of the marker corners. Yoon and Salinas [28,62] proposed offline methods for marker mapping (create a map of squared planar markers) and dealt with the ambiguity problem.

SPM-SLAM [16] builds a marker map incrementally as video frames are captured. However, in this method, it is not possible to correct the errors in the poses of the markers during the process of map creation, so the process must be repeated with new images.

As an improvement over SPM-SLAM, UcoSLAM [17] is introduced. This approach combines natural keypoints and squared fiducial markers. An advantage of the method is that camera localization can be done in the correct scale by observing a single marker. The main drawback of this approach is that the accuracy decreases when no markers are present.

In addition to the methods mentioned above, commercial motion capture systems (Vicon, Optitrack, PTI Visualeyez, etc.) are available on the market to obtain an accurate location of a robot. These systems are usually highly accurate but are very costly, and their use is limited to a certain workspace size. Additionally, these systems require a tedious calibration procedure that needs to be repeated frequently to maintain accuracy.

Our work is an extension of Salinas’ work [28]; however, there are some strategical differences between that approach and ours. First, we use novel smart markers to obtain an accurate map of markers. Second, our proposal seeks to obtain an accurate camera localization by optimizing and reducing errors in the marker map creation process. Furthermore, we remark that all contents of this paper represent a contribution to the community that works with visual odometry and SLAM. Additionally, to date, there are no other similar works in the vision and robotics literature that employ fiducial markers with other sensors in the same encapsulation to enable fusion between visual and IMU/distance data. Most methods focus on equipping a camera and the robot (in any terrestrial, AUV, and UAV) with other sensors but not the marker itself.

## 3. Problem Formulation

In this paper, we deal with two problems, offline mapping and online camera localization. The mapping problem consists of computing a map in the form of a network of 3D rigid transformations, each of them attached to a smart marker (i.e., the graph after optimization). We have opted to model the mapping problem as a pose graph as will be described in Section 3.1.

In general, the offline mapping process has the steps shown in Figure 1. Each of these steps is detailed below:(a)Calibrate a camera;(b)Place a set of smart markers in a scene and capture a video;(c)Detect and get the visual information of each marker (red points referenced to G(0,0,0));(d)Compute the markers’ poses using the points obtained in step (c);(e)Compute relative transforms between all pairs of markers’ poses;(f)Measure the markers points (blue points) and poses (referenced to G(0,0,0)) using the PMS unit of each marker;(g)Compute relative transforms between all pairs of measured markers’ poses;(h)Using the markers’ poses of step (d), the measured markers’ poses of step (f), and the relative transformations of steps (e) and (g), create a graph of markers’ poses and optimize it;(i)After optimization save the optimal markers points (green points) and poses to use it in a online camera localization process.

The second problem covered in this section is the camera localization using an optimal marker map. In the same way as for the mapping problem, we describe the problem as subgraphs of poses given by the camera and visible markers, as will be detailed in Section 3.2.

The online camera localization process has the steps shown in Figure 2. Each of these steps is detailed below:(a)Set up camera and robot;(b)Place a set of smart markers in a scene and capture an image;(c)Load the optimized markers’ poses, from the mapping stage;(d)Compute the camera pose captured by the fist visible marker;(e)Compute the camera poses captured by the rest of visible markers (in case of existing) in the image;(f)Using the markers’ poses of step (c), the camera pose of step (d), and the additional poses of step (e), create a graph of camera poses and optimize it.

### 3.1. Marker Mapping

In this section, we formulate mathematically and represent in a pose graph the mapping problem. Figure 3 shows two markers referenced in the same coordinate system, the global reference system *G*. We can also see that each marker has four points (artificial landmarks) $p_{1}\left(x_{1},y_{1},z_{1}\right)$, $p_{2}\left(x_{2},y_{2},z_{2}\right)$, $p_{3}\left(x_{3},y_{3},z_{3}\right)$ and $p_{4}\left(x_{4},y_{4},z_{4}\right)$, with respect to G(0,0,0).

Using these four points we can compute the unit vectors x, y, z that define the local coordinate system attached to the marker plane and the normalized vectors $\hat{x}$, $\hat{y}$, $\hat{z}$ with Equations (Equation 1) and (Equation 2), respectively.(1)$$\begin{array}{ccc} x=p_{2}-p_{1}, & y=p_{1}-p_{4}, & z=x\times y \end{array}$$
(2)$$\begin{array}{c} \hat{x}=\frac{x}{\left||x|\right|}=\left({\hat{x}}_{1},{\hat{x}}_{2},{\hat{x}}_{3}\right),\\ \hat{y}=\frac{y}{\left||y|\right|}=\left({\hat{y}}_{1},{\hat{y}}_{2},{\hat{y}}_{3}\right),\\ \hat{z}=\frac{z}{\left||z|\right|}=\left({\hat{z}}_{1},{\hat{z}}_{2},{\hat{z}}_{3}\right). \end{array}$$

The 3D marker position is defined by Equation (Equation 3), where $t_{x},t_{y}$ and $t_{z}$ are the components of translation vector t measured from G(0,0,0) to the marker center.
(3)$$\begin{array}{ccc} t_{x}=x_{1}+\left|\frac{\left(x_{2}-x_{1}\right)}{2}\right|, & t_{y}=y_{1}-\left|\frac{\left(y_{4}-y_{1}\right)}{2}\right|, & t_{z}=\frac{\sum_{l=1}^{4}z_{l}}{4} \end{array}$$

Then, the estimated pose $P_{i}$ for the marker *i* in a set with *n* markers can be represented by the homogeneous matrix in Equation (Equation 4).(4)$$P_{i}=\left[\begin{array}{cccc} {\hat{x}}_{1} & {\hat{y}}_{1} & {\hat{z}}_{1} & t_{x}\\ {\hat{x}}_{2} & {\hat{y}}_{2} & {\hat{z}}_{2} & t_{y}\\ {\hat{x}}_{3} & {\hat{y}}_{3} & {\hat{z}}_{3} & t_{z}\\ 0 & 0 & 0 & 1 \end{array}\right],\;\;\;\;\;\;\mathrm{where}\;i=1,\ldots ,n.$$

Relying on the fact that with the support of some sensors (for example a PMS unit) we obtain measurements $Q_{i}$ for the poses of each marker present in the set, each $Q_{i}$ is a matrix composed of a rotation matrix (3×3) and a translation vector (3×1); i.e., it is a complete observation of the pose $P_{i}$.

We can fuse the estimated pose $P_{i}$ with the measured pose $Q_{i}$ to obtain an optimal pose for each marker in a map.

With this, our mapping problem becomes a pose fusion problem. One of the ways to represent a fusion problem is using a pose graph [23]. This type of representation is composed of vertices (or nodes) and edges.

Specifically, for our fusion problem, the vertices are the estimated markers’ poses and the edges are observations of the state of each vertex (i.e., poses measured from non-visual sensors) from which the error between the estimated and measured pose is computed.

In Figure 4 the fusion problem is represented in pose graph form. Blue circles represent the vertices $P_{i}$. Additionally, we can see that in the graph there are two types of edges; the $Q_{i}$ (black arrows) are observations of sensors for the markers’ poses, and $T_{i,k}$ (black dotted arrows) is the total number of observations—m=1,…,n(n−1)/2—between two markers’ poses obtained from the smart markers (Equation (Equation 5)).
(5)$$\begin{array}{c} T_{i,k}=Q_{i}^{-1}Q_{k}=\left[\begin{array}{cc} R_{i,k} & t_{i,k}\\ 0 & 1 \end{array}\right],\;\;\;\;\;\;\mathrm{where}\;k\neq i=1,\ldots ,n. \end{array}$$

The error in each type of observation is calculated differently. Equation (Equation 6) is used to compute the error in $Q_{i}$, and Equation (Equation 7) is employed for $T_{i,k}$ observations. $E_{i,k}$ is the total number of observations between two markers’ poses obtained from visual information.
(6)$${}^{u}e_{i}=P_{i}^{-1}Q_{i},\;\;\;\;\;\;\mathrm{where}\;i=1,\ldots ,n.$$
(7)$${}^{b}e_{i,k}={\left(P_{i}^{-1}P_{k}\right)}^{-1}\left(Q_{i}^{-1}Q_{k}\right)=E_{i,k}^{-1}T_{i,k},\;\;\;\;\;\;\mathrm{where}\;k\neq i=1,\ldots ,n.$$

The fusion problem can be expressed as in Equation (Equation 8). In this equation, $P_{1\ldots n}^{*}$ is the pose $P_{i}$ after optimization, *c* is the set of all pairs of markers, and the superscript ⊤ is the transpose. ${}^{u}\Omega_{i}$ represents unary information matrices (for the marker *i*) detailed later in the Section 4.1.2.
(8)$$P_{1\ldots n}^{*}=\underset{P_{1\ldots n}}{\mathrm{argmin}}\sum_{i}^{n}{}^{u}e_{i}^{⊤}{}^{u}\Omega_{i}{}^{u}e_{i}+\sum_{i,k\in c}^{n}{}^{b}e_{i,k}^{⊤}{}^{b}\Omega_{i,k}{}^{b}e_{i,k}$$

The information matrix for the binary edges ${}^{b}\Omega_{i,k}$ quantifies the information of an observation (transformation between two marker poses). Therefore, the more accurate the observation is, the higher the values in the information matrix. This measurement also gives a reliability factor for each edge; i.e., edges with higher information matrices will have a stronger influence on the optimization process.

In order to compute the binary edges, the four 3D points from a marker are transformed with Equation (Equation 9). Then, we compute the mean of errors $\mu_{i}$ between the points ${}^{i}p_{l}$ and their transformation ${}^{i}{\hat{p}}_{l}$ using Equation (Equation 10), for N=4. The variance of errors $\sigma_{i}^{2}$ is obtained with Equation (Equation 11). Finally, the matrix information ${}^{b}\Omega_{i,k}$ is computed by Equation (Equation 12) where *I* is a (6×6) identity matrix.
(9)$${}^{i}{\hat{p}}_{l}=R_{i,k}{}^{i+1}p_{l}+t_{i,k},\;\;\;\;\;\;\mathrm{where}\;l=1,\ldots ,4.$$
(10)$$\mu_{i}=\frac{1}{N}\sum_{l=1}^{4}{}^{i}{\hat{p}}_{l}-{}^{i}p_{l}$$
(11)$$\sigma_{i}^{2}=\frac{1}{N}\sum_{l=1}^{4}{\left(\left({}^{i}{\hat{p}}_{l}-{}^{i}p_{l}\right)-\mu_{i}\right)}^{2}$$
(12)$${}^{b}\Omega_{i,k}=\frac{1}{\sigma_{i}^{2}}I,\;\;\;\;\;\;\mathrm{where}\;k\neq i=1,\ldots ,n.$$

### 3.2. Camera Localization

The camera localization problem aims to compute the camera pose relative to the reference system defined by a set of markers of known poses. For this, it is assumed that at least one marker is present in an image and that the camera intrinsic parameters [26] are known. In essence, solving camera localization involves computing a rigid transformation relating the camera reference frame to the reference frame of a known set of 3D points, as is the case for the perspective-n-point problem [27] or space resection problem [7]. However, in our method, the camera pose is obtained considering the (previously optimized) 3D pose of the smart markers instead of considering 3D points.

We propose that the natural points of the scene are not used to compute the camera localization to avoid errors (due to tracking and correspondence mainly). Then, if a camera views at least one marker of the smart marker map (whose optimal global poses are results of the Equation (Equation 8)), we can compute a camera pose.

In our case, the camera may visualize more than one marker in a single image. In this situation, it is necessary to calculate the camera localization by minimizing the squared error of the poses obtained by each visible marker.

We formulate the camera localization problem as shown in the pose graph in Figure 5. The black arrows represent the observations $C_{i,j}$ of the camera pose given by each visible marker $M_{i}$ (if there are *n* markers visible in the image, there are *n* edges or hypotheses for the camera pose). The red circles represent the vertices for the camera poses $C_{j}$ (to be optimized). The surrounding blue circles represent the vertices for the markers’ poses $P_{i}^{*}$ (measured relative to G(0,0,0)).

Then, the edge error $e_{i,j}$ can be computed with Equation (Equation 13). In this equation, $C_{j}$ is the vertex that is being updated in the optimization loop and $P_{i}^{*}$ vertex is the pose of the marker $M_{i}$, which is fixed (not optimizable).
(13)$$e_{i,j}=C_{i,j}^{-1}\left(P_{i}^{*-1}C_{j}\right),\;\;\;\;\;\;\mathrm{where}\;i=1,\ldots ,n;\;\;\;\;\;j=1,\ldots ,s;$$

Once we define the edges and vertices of our pose graph, we can define the camera localization problem like Equation (Equation 14).
(14)$$C_{1\ldots j}^{*}=\underset{C_{1\ldots j}}{\mathrm{argmin}}\sum_{i,j}e_{i,j}^{⊤}\Omega_{i,j}e_{i,j}$$

Finally, it is necessary to mention that each $C_{j}$ is being optimized online and independently of the other camera poses. In other words, the camera trajectory is optimized for each instant of time *j*.

## 4. Proposed Solution

Solving Equation (Equation 8) means estimating the optimal position of the markers in a map. To solve this problem, we need to define how to obtain the markers’ poses (estimated $P_{i}$) and their measurements $Q_{i}$, which is described next.

### 4.1. Solving Mapping

The $P_{i}$ poses can be obtained from the visual information of simple fiducial markers and the $Q_{i}$ poses can be measured using sensors. To implement this, we decide to attach a square fiducial marker to a PMS unit, creating a new type of artificial marker as described in Section 4.1.1.

#### 4.1.1. Smart Markers

This section describes the first of our contributions, which is a new concept of marker that we named a smart marker. Each marker of this type, in addition to a binary visual pattern, has an inertial measurement unit (IMU) and two ToF distance meters encapsulated in a device that we call PMS unit, as shown in Figure 6.

More specifically, each PMS unit has one IMU LSM9DS1 and two ToF ranging sensors VL53L1X. The typical zero level offset for each sensor in the IMU (when it is static) is ±90 mg for the accelerometer measurement range of ±8 g. For the magnetometer, the offset is ±1 gauss for a range of ±4 gauss. In the gyroscope, the offset is ±30 dps for a range of ±2000 dps [63]. On the other hand, the ToF sensors have a maximum ranging error of ±20 mm in ambient light conditions and when they are configured in a short-distance mode (maximum distance 130 cm) [64]. In addition to the reported values, we performed an accuracy analysis of all sensors used in our approach, with results presented in Section 5.1.

In Figure 7 we can see some important aspects regarding our smart markers. As the inertial navigation reference system is in the IMU chip center, we align and fix it to the IMU, with the marker center *S*. The positive rotation of the marker is defined as clockwise in the direction of the axis of rotation (right-hand rule).

We employ a quaternion representation to describe the marker orientation in a three-dimensional space. The quaternion (obtained from the IMU) that describes the orientation of frame *S* relative to frame *G* is defined by Equation (Equation 15), where $q_{w}$, $q_{x}$, $q_{y}$ and $q_{z}$ are scalars known as quaternion components [65].
(15)$${}_{S}^{G}q=\left[q_{w},q_{x},q_{y},q_{z}\right]$$

The 9-DoF IMU data of a PMS unit are fused using the Madgwick algorithm [66] that runs in an Esp32-Wrover module. In addition to being open-source, this algorithm has the best performance (low computational load and operates at small sampling rates) for embedded systems. The PMS algorithm is implemented with the Arduino IDE. Additionally, the Esp32 module allows the marker to sends its position and orientation in a wireless and synchronized way. It is important to indicate that smart markers can be implemented with similar ICs and sensors to those mentioned before, not necessarily the same.

The marker localization referenced to *G* can be represented using directly the relative distance or the translation components. The relative distance between two markers is the square root of the sum of the squares of the *x* and *y* translation components. Then, if the measurements of the components have errors (no matter how small these are), the relative distance will have even bigger errors. For this reason, we use the individual translation components to represent the marker localization.

In Figure 7 we can see a map with four markers placed using a previous alignment i.e., placed in a parallel (same plane xy) or co-planar form to the marker $M_{1}$. The marker $M_{1}$ is considered the principal marker of the map because in their center is the global coordinate system G(0,0,0).

Then, the relative translation components of the $M_{2}$ referenced to *G* can be measured using only two ToF sensors. One sensor measures the translation ${\overset{˘}{t}}_{z(2,1)}$ along its *z* axis and the other sensor measures the relative translation ${\overset{˘}{t}}_{x(2,3)}$ along its *x* axis. In the case of the $M_{n}$ the localization referenced to *G* is the sum of the ${\overset{˘}{t}}_{z(n,2)}$ and ${\overset{˘}{t}}_{z(2,1)}$.

Is necessary to take into consideration that the markers must be placed in the same plane xz, that is, at a similar height on the *y* axis. This restriction occurs because ToF sensors need at least 50% of a Region of Interest (ROI) to be seen to accurately measure a distance [67].

Finally, each PMS unit measures the pose $Q_{i}$ for the marker *i* in a set with *n* markers, using the homogeneous matrix in Equation (Equation 16).
(16)$$\begin{array}{c} Q_{00}=2q_{w}^{2}+2q_{x}^{2}-1\\ Q_{01}=2\left(q_{x}q_{y}-q_{w}q_{z}\right)\\ Q_{02}=2\left(q_{x}q_{z}+q_{w}q_{y}\right)\\ Q_{10}=2\left(q_{x}q_{y}+q_{w}q_{z}\right)\\ Q_{11}=2q_{w}^{2}+2q_{y}^{2}-1\\ Q_{12}=2\left(q_{y}q_{z}-q_{w}q_{x}\right)\\ Q_{20}=2\left(q_{x}q_{z}-q_{w}q_{y}\right)\\ Q_{21}=2\left(q_{y}q_{z}+q_{w}q_{x}\right)\\ Q_{22}=2q_{w}^{2}+2q_{z}^{2}-1\\ Q_{i}=\left[\begin{array}{cccc} Q_{00} & Q_{01} & Q_{02} & {\overset{˘}{t}}_{x}\\ Q_{10} & Q_{11} & Q_{12} & {\overset{˘}{t}}_{y}\\ Q_{20} & Q_{21} & Q_{22} & {\overset{˘}{t}}_{z}\\ 0 & 0 & 0 & 1 \end{array}\right],\;\;\;\;\;\;\mathrm{where}\;i=1,\ldots ,n. \end{array}$$

#### 4.1.2. PMS Accuracy Representation

Our proposed PMS unit is made up of low-cost sensors (LSM9DS1 and VL53L1X), and hence, the pose measurements can have a certain level of error. This error can be represented using an asymmetric and positive semi-definite matrix known as the information matrix. More specifically the matrix ${}^{u}\Omega_{i}$ for marker i=1,…,n (Equation (Equation 17)) represents the information of the measurement error and can be computed from a inverse of the covariance matrix ${\mathrm{Cov}}_{i}$.

We assume, to simplify the problem that pose variables (*x*, *y*, *z*, $q_{x}$, $q_{y}$, $q_{z}$) are independent (or uncorrelated), then ${}^{u}\Omega_{i}$ can be represented like a diagonal matrix.
(17)$${}^{u}\Omega_{i}={\mathrm{Cov}}_{i}^{-1}=\left[\begin{array}{cccccc} 1/\sigma_{x}^{2} & 0 & 0 & 0 & 0 & 0\\ 0 & 1/\sigma_{y}^{2} & 0 & 0 & 0 & 0\\ 0 & 0 & 1/\sigma_{z}^{2} & 0 & 0 & 0\\ 0 & 0 & 0 & 1/\sigma_{q_{x}}^{2} & 0 & 0\\ 0 & 0 & 0 & 0 & 1/\sigma_{q_{y}}^{2} & 0\\ 0 & 0 & 0 & 0 & 0 & 1/\sigma_{q_{z}}^{2} \end{array}\right]$$

#### 4.1.3. Mapping Optimization

Solving the mapping problem involves finding a solution to the Equation (Equation 8). One of the ways existing in the literature to solve this equation is by using graph optimization.

The basic idea of graph optimization is to represent nonlinear least-squares problems as embedded graphs and optimize them. This type of problem representation generally is formed for nodes (vertices) and constraints (edges). The optimization aims to find an optimal estimation of the node’s values which minimizes the errors determined by the constraints.

In our implementation, we use the popular graph optimization library g2o [23]. The g2o optimizer can be considered as the optimization engine managing everything. As the optimization problems in mapping and localization (SLAM) fields are commonly mostly sparse, we use the sparse optimizer.

The next step in the g2o optimization process is to define an optimization algorithm. Defining this algorithm involves knowing the update equation for the mapping problem.

In order to define the update equation we can rewrite the Equation (Equation 8) like the Equation (Equation 18), where *v* and *w* represents all unknowns in the unary (Equation (Equation 6)) and binary (Equation (Equation 7)) error respectively.
(18)$$F\left(v,w\right)=\sum_{i}^{n}{}^{u}e_{i}{\left(v\right)}^{⊤}{}^{u}\Omega_{i}{}^{u}e_{i}\left(v\right)+\sum_{i,k\in c}^{n}{}^{b}e_{i,k}{\left(w\right)}^{⊤}{}^{b}\Omega_{i,k}{}^{b}e_{i,k}\left(w\right)$$

Then the optimization problem can be stated as in Equation (Equation 19).
(19)$$v^{*},w^{*}=\underset{v,w}{\mathrm{argmin}}F\left(v,w\right)$$

We solve the Equation (Equation 19) using Gauss–Newton method. If we find $v_{0}$ and $w_{0}$ as the initial/rough estimations of *v* and *w*, the next step would be looking for a $\Delta v^{*}$ and $\Delta w^{*}$ to achieve the Equation (Equation 20).
(20)$$\Delta v^{*},\Delta w^{*}=\underset{\Delta v,\Delta w}{\mathrm{argmin}}F\left(v_{0}+\Delta v,w_{0}+\Delta w\right)$$
where $F(v_{0}+\Delta v,w_{0}+\Delta w)$ can be expanded to be Equation (Equation 21).
(21)$$F\left(v_{0}+\Delta v,w_{0}+\Delta w\right)=\sum_{i}^{n}{}^{u}e_{i}{\left(v_{0}+\Delta v\right)}^{⊤}{}^{u}\Omega_{i}{}^{u}e_{i}\left(v_{0}+\Delta v\right)+\sum_{i,k\in c}^{n}{}^{b}e_{i,k}{\left(w_{0}+\Delta w\right)}^{⊤}{}^{b}\Omega_{i,k}{}^{b}e_{i,k}\left(w_{0}+\Delta w\right)$$

Applying first order Taylor approximation on ${}^{u}e_{i}$ and ${}^{b}e_{i,k}$, we obtain the Equations (Equation 22) and (Equation 23).
(22)$${}^{u}e_{i}\left(v_{0}+\Delta v\right)\approx {}^{u}e_{i}\left(v_{0}\right)+\frac{d({}^{u}e_{i})}{dP_{i}}\Delta v$$
(23)$${}^{b}e_{i,k}\left(w_{0}+\Delta w\right)\approx {}^{b}e_{i,k}\left(w_{0}\right)+\frac{d({}^{b}e_{i,k})}{dD_{i,k}}\Delta w$$

From Equations (Equation 22) and (Equation 23) we can obtain the Jacobian matrices for the unary (Equation (Equation 24)) and binary (Equation (Equation 25)) error respectively. In this last equation, $D_{i,k}=P_{i}^{-1}P_{k}\;\mathrm{for}\;k\neq i=1,\ldots ,n$.
(24)$${}^{u}J_{i}=\frac{d({}^{u}e_{i})}{dP_{i}}$$
(25)$${}^{b}J_{i,k}=\frac{d({}^{b}e_{i,k})}{dD_{i,k}}$$

With these Jacobian matrices, we can get the Equation (Equation 26).
(26)$$\begin{array}{c} F\left(v_{0}+\Delta v,w_{0}+\Delta w\right)\approx \sum_{i}^{n}{\left({}^{u}e_{i}\left(v_{0}\right)+{}^{u}J_{i}\Delta v\right)}^{⊤}{}^{u}\Omega_{i}\left({}^{u}e_{i}\left(v_{0}\right)+{}^{u}J_{i}\Delta v\right)+\\ \sum_{i,k}^{n}{\left({}^{b}e_{i,k}\left(w_{0}\right)+{}^{b}J_{i,k}\Delta w\right)}^{⊤}{}^{b}\Omega_{i}\left({}^{b}e_{i}\left(w_{0}\right)+{}^{b}J_{i}\Delta w\right)\\ =\sum_{i}^{n}\left\{{}^{u}e_{i}{\left(v_{0}\right)}^{⊤}{}^{u}\Omega_{i}{}^{u}e_{i}\left(v_{0}\right)+\left[2{}^{u}e_{i}{\left(v_{0}\right)}^{⊤}{}^{u}\Omega_{i}{}^{u}J_{i}\left(v_{0}\right)\right]\Delta v+\Delta v^{⊤}\left[{}^{u}J_{i}{\left(v_{0}\right)}^{⊤}{}^{u}\Omega_{i}{}^{u}J_{i}\left(v_{0}\right)\right]\Delta v\right\}+\\ \sum_{i,k}^{n}\left\{{}^{b}e_{i,k}{\left(w_{0}\right)}^{⊤}{}^{b}\Omega_{i,k}{}^{b}e_{i,k}\left(w_{0}\right)+\left[2{}^{b}e_{i,k}{\left(w_{0}\right)}^{⊤}{}^{b}\Omega_{i,k}{}^{b}J_{i,k}\left(w_{0}\right)\right]\Delta w+\Delta w^{⊤}\left[{}^{b}J_{i,k}{\left(w_{0}\right)}^{⊤}{}^{b}\Omega_{i,k}{}^{b}J_{i,k}\left(w_{0}\right)\right]\Delta w\right\} \end{array}$$

Simplifying Equation (Equation 26) we get:(27)$$\begin{array}{c} F\left(v_{0}+\Delta v,w_{0}+\Delta w\right)\approx \sum_{i}^{n}a_{i}\left(v_{0}\right)+2b_{i}\left(v_{0}\right)\Delta v+\Delta v^{⊤}H_{i}\left(v_{0}\right)\Delta v+\\ \sum_{i,k}^{n}a_{i,k}\left(w_{0}\right)+2b_{i,k}\left(w_{0}\right)\Delta w+\Delta w^{⊤}H_{i,k}\left(w_{0}\right)\Delta w \end{array}$$
where $a_{i}\left(v_{0}\right)$, $b_{i}\left(v_{0}\right)$, $H_{i}\left(v_{0}\right)$, $a_{i,k}\left(w_{0}\right)$, $b_{i,k}\left(w_{0}\right)$ and $H_{i,k}\left(w_{0}\right)$ are defined from the Equations (Equation 28) to (Equation 33), respectively.
(28)$$a_{i}\left(v_{0}\right)={}^{u}e_{i}{\left(v_{0}\right)}^{⊤}{}^{u}\Omega_{i}{}^{u}e_{i}\left(v_{0}\right)$$
(29)$$b_{i}\left(v_{0}\right)={}^{u}e_{i}{\left(v_{0}\right)}^{⊤}{}^{u}\Omega_{i}{}^{u}J_{i}\left(v_{0}\right)$$
(30)$$H_{i}\left(v_{0}\right)={}^{u}J_{i}{\left(v_{0}\right)}^{⊤}{}^{u}\Omega_{i}{}^{u}J_{i}\left(v_{0}\right)$$
(31)$$a_{i,k}\left(w_{0}\right)={}^{b}e_{i,k}{\left(w_{0}\right)}^{⊤}{}^{b}\Omega_{i,k}{}^{b}e_{i,k}\left(w_{0}\right)$$
(32)$$b_{i,k}\left(w_{0}\right)={}^{b}e_{i,k}{\left(w_{0}\right)}^{⊤}{}^{b}\Omega_{i,k}{}^{b}J_{i,k}\left(w_{0}\right)$$
(33)$$H_{i,k}\left(w_{0}\right)={}^{b}J_{i,k}{\left(w_{0}\right)}^{⊤}{}^{b}\Omega_{i,k}{}^{b}J_{i,k}\left(w_{0}\right)$$

If F in the Equation (Equation 27) is convex, or at least locally convex in the region close to $\left(v_{0},w_{0}\right)$, $F(v_{0}+\Delta v,w_{0}+\Delta w)$ is the minimum if the gradient ∇F is equals to 0 (Equation (Equation 34)).
(34)$$\nabla F=\left[\begin{array}{c} \frac{\partial F}{\partial \Delta v}\\ \frac{\partial F}{\partial \Delta w} \end{array}\right]=0$$
which leads to:(35)$$\left[\begin{array}{c} \sum_{i}^{n}\left[2b_{i}\left(v_{0}\right)+2H_{i}\Delta v^{*}\right]\\ \sum_{i,k}^{n}\left[2b_{i,k}\left(w_{0}\right)+2H_{i,k}\Delta w^{*}\right] \end{array}\right]=0$$
then we can solve $\Delta v^{*}$ and $\Delta w^{*}$:(36)$$\left[\begin{array}{c} \sum_{i}^{n}H_{i}\left(v_{0}\right)\Delta v^{*}\\ \sum_{i,k}^{n}H_{i,k}\left(w_{0}\right)\Delta w^{*}] \end{array}\right]=\left[\begin{array}{c} -\sum_{i}^{n}b_{i}\left(v_{0}\right)\\ -\sum_{i,k}^{n}b_{i,k}\left(w_{0}\right) \end{array}\right]$$

The above equation can be arranged in a sparse matrix in the form of the Equation (Equation 37). This equation is known as the update optimization equation.
(37)$$\left[\begin{array}{c} H\left(v_{0}\right)\Delta v^{*}\\ H\left(w_{0}\right)\Delta w^{*} \end{array}\right]=\left[\begin{array}{c} -b(v_{0})\\ -b(w_{0}) \end{array}\right]$$

In order to find $\Delta v^{*}$ and $\Delta w^{*}$, its necessary to compute $H{\left(v_{0}\right)}^{-1}$ and $H{\left(w_{0}\right)}^{-1}$. Just as mentioned above, we assume that *H* is a sparse matrix and its inverse can be computed using a g2o sparse solver.

The g2o solver needs the method for solving the inverse matrix and the sparse matrix structure to be defined. Since our problem is of type Ax=b, we use a linear solver, and Cholmod [68] as the back-end library to solve this linear problem.

When solving for a sparse matrix, g2o treats the matrix as blocks to do the Schur Complement. To treat it correctly it is necessary to specify how the matrix is structured. For the mapping problem, we optimize the 6-DoF marker pose; therefore, the block size of the matrix *H* will be 6×1.

Once we specific the g2o solver, we can define the optimization algorithm. To our implementation, we use the Gauss–Newton algorithm.

After that the optimization algorithm is fully defined and created, we can model the optimization problem by adding vertices and edges into the g2o optimizer. Here we define the reference marker vertex as fixed and the other as optimizable. Finally, after running the optimizer for some interactions we can get the optimal (in the least-squares sense) result for the Equation (Equation 8) as shown in the experiments in Section 5.3.

### 4.2. Solving Camera Localization

In Section 3.2 we define the camera localization problem like the Equation (Equation 14). The solution to this problem involves the use of the optimal smart marker map described in Section 4.1.1.

Then, in the same form that in the previous section we solve the camera localization problem using a pose graph optimization with g2o.

#### Optimization of Camera Localization

In order to find the update equation for the camera localization problem we can rewrite the Equation (Equation 14) like the Equation (Equation 38), where *s* represents all unknowns in the error defined for the Equation (Equation 13).
(38)$$C\left(s\right)=\sum_{i}^{n}e_{i}{\left(s\right)}^{⊤}\Omega_{i}e_{i}\left(s\right)$$

Solving the Equation (Equation 38) using the same process that was shown in Section (Section 4.1.3) we can find the Equation (Equation 39) that is the optimization update equation for the camera problem.
(39)$$H\left(s_{0}\right)\Delta s^{*}=-b\left(s_{0}\right)$$

To estimate $\Delta s^{*}$ it is necessary to compute the inverse of *H*. Then, to calculate this inverse we assume that *H* is a sparse matrix and use a g2o sparse solver. Specifically, we use a solver of type linear and a block size for *H* of 6×1 because we are optimizing a 6 DoF camera pose.

We use the Gauss–Newton algorithm to optimize and adding vertexes and edges into the g2o optimizer. Here the vertices for the markers’ poses are fixed and a camera vertex is optimizable.

After running the optimizer for some interactions we can find the optimal camera localization as shown in the experiments in Section 5.4.

Finally, note that for the mapping implementation in Section 4.1, it is necessary that a frame has at least two visible markers for the optimization process to be computed. After that, the camera localization of the Section 4.2 can be estimated even if only one marker is visible in a frame.

## 5. Experiments

In this section, we detail the experiments carried out to validate our proposal. First, in Section 5.1 we evaluate the accuracy of the orientation and position sensors. LSM9DS1 and PCEVDL16I sensors were used to measure the markers and camera orientation, respectively. The VL53L1X is used to measure the PMS localization within a map and the GLM80 was used to measure the camera position.

Since our method is built upon two different processing stages (mapping and camera localization), we conducted two different experiments, each of them being responsible for assessing a different stage of the system. The first experiment evaluates the creation process of the map of smart markers, i.e., the graph with the optimized poses of the smart markers. For this, two different datasets are used. The second experiment evaluates the camera localization stage in two distinct datasets. In both experiments, the datasets cover different configurations of markers placed throughout the environment.

Our method is evaluated quantitatively by two different error types: Relative Positioning Error (RPE) and Absolute Trajectory Error (ATE). With these two measurements, it is possible to highlight the capabilities of the proposed system concerning different properties.

The RPE metric is used to quantify the accuracy of the built map of smart markers by measuring the relative transformations between the poses of all possible pairs of smart markers in comparison to the ground truth. This is especially useful if only sparse, relative relations are available as ground truth [69], which is our case. Besides, the translational or rotational counterparts of this metric can be evaluated separately. A lower RPE error is evidence that the spatial relations between the smart markers are consistent with the actual placement of the markers on the environment.

On the other hand, the global consistency of a camera trajectory is evaluated using the ATE metric. In this error type, the error is computed directly between the estimated camera poses and a corresponding ground truth camera pose.

### 5.1. Evaluation of Sensors

Our approach uses two ground truths, one for the markers’ poses and the other for the camera poses. These two ground truths were made by hand using a modified (to capture data at a rate of 15 Hz) laser distance meter Bosh GLM80 and a vibration analyzer PCEVDL16I. Moreover, a smart marker has a square fiducial marker and a PMS unit. Each PMS unit has one 9-DoF IMU LSM9DS1 and two ToF ranging sensor VL53L1X.

We set up the accelerometers of LSM9DS1 and PCEVDL16I, with a measurement range of ±2 g and ±16 g, respectively. While that LSM9DS1 gyroscope was set up with a range of ±245 dps.

The sensor VL53L1X was configured to use a field of view of 27${}^{\circ }$ and a short-distance mode (min. dist. 4 cm, max. dist. 130 cm).

In this section, we analyze all sensors in the same conditions, in a laboratory with a temperature of 22 ${}^{\circ }$C, ambient light of ≈350 lux, and a capture rate of 15 Hz.

We use the Equations (Equation 40)–(Equation 42), to compute the accuracy, repeatability and total error, respectively.

The accuracy $a_{i}$ is the difference between the mean of $m_{j=1,\ldots ,k}$ measures and the actual measure $m_{i=1,\ldots ,l}$.
(40)$$a_{i}=\mu \left(m_{j}\right)-m_{i}=\frac{1}{k}\sum_{j=1}^{k}m_{j}-m_{i}$$

The repeatability $r_{i}$ is the standard deviation of the mean of $m_{j=1,\ldots ,k}$ measurements.
(41)$$r_{i}=\frac{\sigma_{i}}{\sqrt{k}}=\frac{1}{k}{\left(\sum_{j=1}^{k}|m_{j}-\mu \left(m_{j}\right){|}^{2}\right)}^{1/2}$$

The total error $t_{i}$ of the actual measure $m_{i=1,\ldots ,l}$ is the sum of accuracy $a_{i}$ and repeatability $r_{i}$.
(42)$$t_{i}=a_{i}+r_{i}$$

#### 5.1.1. Evaluation of Orientation Sensors

The analysis was made to verify the accuracy, repeatability, and total error in terms of degrees (roll, pitch, yaw) from each axis when the sensor was static with no physical acceleration or magnetic fields introduced.

The data for the orientation sensors analysis was obtained using an automatic mobile platform and an LSM9DS1 sensor (with Madgwick fusion algorithm [66]), as shown in Figure 8. For each axis of rotation (x,y,z) we take $m_{i=0,\ldots ,90^{\circ }}$ angles measures. The interval or step in between each set of measures $m_{i}$ is one degree.

We capture data for five minutes, i.e., we get k=4500 measurements for each angle between 0 to 90${}^{\circ }$. The results of accuracy, repeatability and total error for *x*–roll, *y*–pitch and *z*–yaw axes of the LSM9DS1 sensor, are show in the Figure 9a–c, respectively.

In the same form, using the device PCEVDL16I we take $m_{j=1,\ldots ,4500}$ measurements for $m_{i=0,\ldots ,90^{\circ }}$ angles with a step of 1${}^{\circ }$. In the Figure 10a–c we can see the results for this device.

#### 5.1.2. Evaluation of Position Sensors

The data for the distance sensors evaluation was obtained using a VL53L1X sensor and a grey 17% reflectance chart target, as shown in Figure 11. We maintain the sensor static and move the target from 4 to 130 cm, then a set of $m_{j=1,\ldots ,k}$ distance measurements is taken each centimeter.

After five minutes of samples for each distance $m_{i}$, we get a total of k=4500 measures, then we compute the accuracy $a_{i}$, repeatability $r_{i}$, and total error $t_{i}$ for the sensor configured in short mode.

In Figure 12a,c,e, we can see the results of the analysis for the VL53L1X.

In the same form, using the laser meter GLM80 we take $m_{j=1,\ldots ,4500}$ measurements for $m_{i=10,\ldots ,1500\;\mathrm{cm}}$ distances with a step of 10 cm. In the Figure 12b,d,f we can see the results for the laser meter used to create the ground truths of the markers and camera position.

Finally, for better understanding, the graphics results, the limits of accuracy, repeatability, and total error for each sensor analyzed are presented in Table 1.

### 5.2. Hardware Setup and Dataset Description for Mapping and Camera Localization Experiments

All experiments were run on an nVidia Jetson Xavier. The ArUco library [19] was employed for marker detection in the video sequences recorded with a calibrated Stereolabs ZED camera [70,71] mounted on top of a Pioneer 3AT robot, which is shown in Figure 13. All sensors and devices were calibrated and shared the same coordinate system. The measures taken by any device in the platform are referenced to the left camera coordinate system, as shown in Figure 13.

Using the ZED camera we record four datasets for the experiments. Table 2 provides details about the recorded datasets. The sequences **M-Seq1/hall** and **T-Seq3/hall** have fourteen (30 × 30 cm) smart markers placed in an indoor hall of size 2 × 20 m (Figure 14a). The sequences **M-Seq2/room** and **T-Seq4/room** consists of images capturing nine smart markers distributed within a rectangular room of size 7 × 5 m (Figure 15a).

### 5.3. Evaluation of Smart Marker Mapping

In this section, we discuss the results of the experiments assessing the accuracy of the maps built by our method. The mapping stage is evaluated using the datasets **M-Seq1/hall** and **M-Seq2/room**. To do this, we compare the Marker Mapper [28] method from the ArUco library to our method using the poses of all markers as ground truth. The mapping results are shown in Figure 14 and Figure 15.

For the experiment with the dataset **M-Seq1/hall** (Figure 14c), specifically in the *x* translation component, we can see that the two approaches are close to ground truth until marker #5. A slight difference between the results of our method and Marker Mapper can be seen starting from marker #6. In the *y* component is where the greatest difference between the two proposals occurs. Our last marker #13 is at y=−5 cm approximately, while the estimate from Marker Mapper has y=29 cm. These differences can be seen more clearly in the 3D reconstruction shown in Figure 15b. In the *z* translation component, both our method and Marker Mapper are very close to ground truth and do not exhibit large errors.

The results for the experiment with the dataset **M-Seq2/room** are shown in Figure 15c. In the *x* component, the two approaches are close to the ground truth. However, for the *y* component, the estimates of the Marker Mapper method have markers #1 to #8 with large errors (2 to 6 centimeters) for the height of the markers compared to the ground truth. In *z* it can be seen that all markers pose on our map have a small error regarding the ground truth while the Marker Mapper poses #5 to #7 have the largest errors. This fact can be better visualized in Figure 14b.

Table 3 shows the translational and rotational RMSE of RPE quantifying the accuracy of our method and Marker Mapper method. As can be observed from the results, our method is more accurate in both datasets compared to Marker Mapper. More importantly, the accuracy improvement is substantial, since our method can reduce the errors by one order of magnitude for the translational RPE in the two datasets. Specifically, for the dataset **M-Seq1/hall**, our map has a translational RPE 86% less than the Marker Mapper map. For the dataset **M-Seq2/room**, our approach reduces the error in 88% in comparison to the Marker Mapper map.

Table 3 also shows the processing times for the mapping stage of our method and Marker Map method. The values presented in the table are similar for both approaches. Nevertheless, this stage is performed offline by both methods and hence, the processing times are shown to provide an evidence for the fact that the use of smart markers does not incur larger computation times.

### 5.4. Evaluation of Camera Localization Using Smart Markers

In this second experiment, we discuss the accuracy of the online stage (camera localization) of the proposed method. For this, we capture images of smart markers with known poses (computed from the mapping stage). The camera poses computed by the camera localization stage of our method are compared to camera poses computed by UcoSLAM [17], LibViso2 [29] and the proprietary ZED tracking API [30]. The first approach is a keypoint (artificial and/or natural) based SLAM method able to detect squared fiducial markers, and thus allowing scale estimation for the monocular SLAM case. The second approach uses an 8-point algorithm for fundamental matrix estimation and estimates scale assuming that the camera is moving at a known and fixed height above the ground. Finally, the third method uses stereo vision to estimate the camera 6-DoF pose with a proprietary algorithm. These last two approaches use natural keypoints to estimate camera pose.

Of the methods mentioned above, the only one that uses artificial keypoints is UcoSLAM; therefore, it is the only one that is in the same category as our proposal. As will be shown later, our method is more accurate. Comparisons with the other two approaches (using natural keypoints) are to demonstrate the accuracy benefits of using smart markers over natural keypoints.

For the evaluation of camera localization, we use datasets **T-Seq3/hall** and **T-Seq4/room**. In these data sequences, the camera odometry trajectories are analyzed in two different locations: on a straight hall and a rectangular room. In both cases, the ground truth trajectories were measured using industrial distance meter laser and IMU.

The accuracy of the computed poses are assessed by the RPE and ATE error metrics. While the ATE provides an absolute comparison to the ground truth pose, the RPE error provides a measurement of local trajectory consistency [72]. We compute the RPE with Δ=1, i.e., we compute the error between consecutive image pairs. For example in the dataset **T-Seq3/hall** we have 750 frames, using Δ=1, we have 749 pairs of frames to compute the RPE.

Figure 16 shows the results for the camera localization evaluation. For dataset **T-Seq3/hall**, Figure 16a shows the comparison of translation components of the trajectories of all evaluated methods and the ground truth.

In the *x* component LibViso2 results is close to ground truth from t=0 to t=35. From t=35 to the end of the trajectory, the trajectory computed by the ZED API is the best of all evaluated methods. Nevertheless, our method is more accurate than UcoSLAM, which also uses artificial markers for localization, throughout all time steps. As far as the *y* component is concerned, we can see that our trajectory is closer to the ground truth than the other three methods evaluated. The same behavior can be observed for the *z* component for our method and UcoSLAM for the whole data sequence.

Camera trajectories for dataset **T-Seq4/room** are shown in Figure 16b. In the *x* coordinate, the trajectory computed by ZED API follows closely the ground truth trajectory, while the trajectory computed by our method and by UcoSLAM are very similar. In the *y* component it is clear that our method is capable of estimating a better approximation of the camera localization than the other three methods. Regarding the *z* coordinate, the ZED API method computes the most accurate trajectory from the beginning of the trajectory until t=32. After this time, the result of our method is more accurate.

In Figure 16a,b, we can visualize the trajectory computed by the camera localization stage of our method along with the artificial markers computed by the mapping stage of our method and by the ArUco marker map method. These figures highlight the fact that a better marker-based trajectory is directly related to the accuracy of the mapping stage i.e., an accurate marker map produces an accurate camera localization.

Table 4 shows more details of a quantitative evaluation of the proposed method. From the results in this table, we can observe that our method gives a trajectory with an ATE of 13.7 cm for dataset **T-Seq3/hall** and 12.4 cm for dataset **T-Seq4/room**. In both cases, this metric is the lowest compared to the results obtained with the other methods under analysis (UcoSLAM, LibViso2, and ZED tracking API). Error reductions are of ≈55% and ≈40% for both datasets compared to the second-best method (UcoSLAM). Similarly, for the translational RPE metric, we can see that the lowest metric is that computed by our method, more specifically 0.4 cm for dataset **T-Seq3/hall** and 1.3 cm for dataset **T-Seq4/room**. In comparison to ZED API, which has the second-best result in **T-Seq3/hall**, our method resulted in error reductions of ≈50% and ≈44% in comparison to LibViso2 for dataset **T-Seq4/room**.

The last column of the Table 4 shows the results of the processing times of all evaluated methods. As can be observed, our method is only marginally slower than LibViso2, the fastest method of all evaluated. We note that camera localization is a task mostly performed in an online processing pipeline, for which the processing times of our method (less than 3 ms per frame) are more than sufficient.

## 6. Discussion

In general, from the experiments carried out, it can be concluded that the accuracy of the marker maps directly influences the camera localization. For example, in the case of UcoSLAM, its marker map (UcoSLAM uses Marker Mapper to create the map) has a large error in the *y* axis; therefore, this error is also reflected in the *y* component of the camera pose translation.

Additionally, we can notice that the large errors in the *y* component are also present in the other two compared approaches. In the case of monocular LibViso2, this occurs because in the form in which the camera poses are computed, this approach uses the camera height above ground and the pitch angle. We notice that the camera pose results are very sensitive to the setup of these two parameters, as minimal errors cause variations in the results. For the Stereolabs API results we notice that the camera poses are very influenced by the resolution and the frame rate. With a large resolution (2208×1242 px) the frame rate decreases (to 15 f.p.s.), which ultimately leads to blurred images and errors in keypoint detection and matching, and consequently, errors in pose computation.

The use of a set of smart markers with optimized inter-marker relative poses makes it possible to obtain accurate camera localization in a second stage performed online. As demonstrated in Section 5, our method estimates the camera height parallel to the floor plane, unlike all other methods, which show noticeable drift in the vertical direction.

## 7. Limitations

In Section 3.1 and Section 3.2 we established that our camera mapping and localization methods depend on the restriction that the smart markers have to be grid aligned (i.e., collinear or coplanar), which can be certainly considered a limitation.

Undoubtedly the use of smart markers may require more implementation time and more money in relation to printing normal fiducial markers. This is especially more costly according to workspace limitations. For example, if an application seeks to track a drone in a space of 100 m${}^{2}$, it is necessary to have a large number of smart markers to cover the area.

In general, if an application seeks accuracy, it can benefit from the use of smart markers.

## 8. Conclusions and Future Work

This paper proposes a method for accurate marker mapping and camera localization using novel smart fiducial markers called smart markers. The method creates an initial map composed of relative poses of the observed markers. Then, these initial poses are fused with PMS measures through the pose graph optimization framework. Finally, with the optimal smart marker map, camera localization tasks can be performed.

We evaluated the marker mapping process and compared our method with the Marker Mapper [28] method. Results demonstrate that our approach decreases the marker map RPE between ≈85% and ≈90%. In the same way, we compared our approach with SLAM methods (LibViso2, UcoSLAM, ZED Tracking API) to evaluate the camera localization process. For this task, results show that our work decreases the ATE by ≈50%.

In this work we have focused on improving accuracy in marker mapping and camera localization by correcting marker pose errors online with the use of PMS units. As future work we propose to create an incremental method—that is, the marker map can be expanded online. Finally, another aspect that we plan to improve is the ability to place the markers in any position. Currently, our method is limited to using smart markers parallel or co-planar to the reference marker.

## Figures and Tables

**Figure 1 sensors-21-00625-f001:**
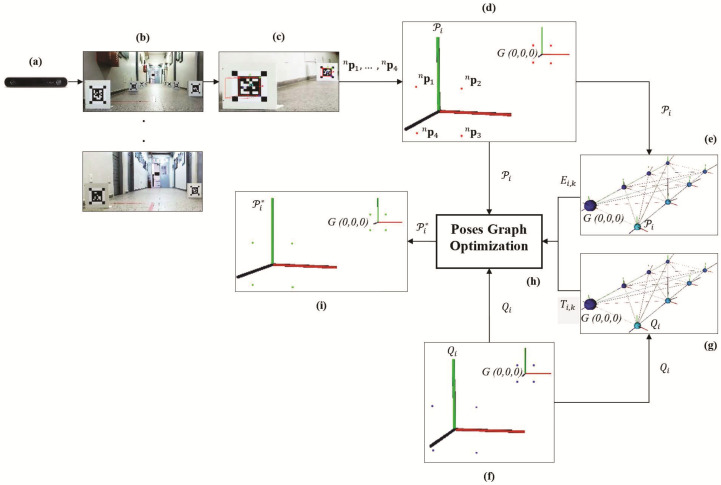
Diagram of steps for the marker map’s construction. (**a**) The first step is to calibrate a camera. (**b**) Then, place a set of smart markers in a scene and capture a video. (**c**) Detect and get the points of each marker. Compute the markers’ poses using the points (**d**), and the relative transforms between all pairs of markers’ poses (**e**). (**f**) Measure the markers’ points and poses using the PMS unit of each marker. (**g**) Compute the relative transforms between all pairs of measured markers’ poses. (**h**) Create a graph of markers’ poses and optimize it. (**i**) Finally, after optimization save the optimal markers points and poses to use in an online camera localization process.

**Figure 2 sensors-21-00625-f002:**
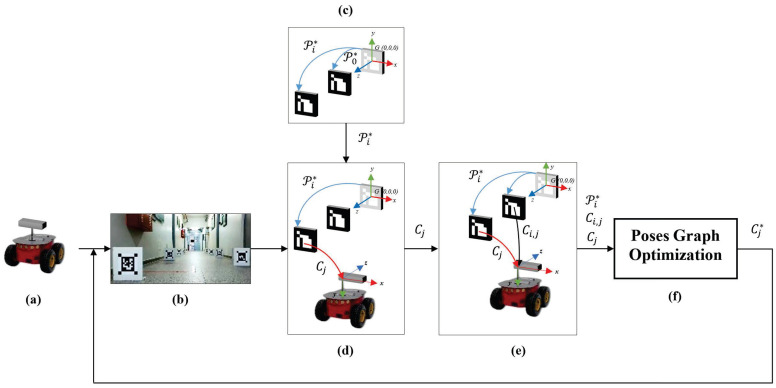
Diagram of steps for the camera localization. (**a**) Set up camera and robot. (**b**) Place a set of smart markers in a scene and capture an image. (**c**) Load the optimized markers’ poses, from the mapping stage. (**d**) Compute the camera pose captured by the fist visible marker. (**e**) Compute the camera poses captured by the rest of visible markers (in case of existing) in the image. (**f**) Using the markers’ poses of step (**c**), the camera pose of step (**d**), and the additional poses of step (**e**), create a graph of camera poses and optimize it.

**Figure 3 sensors-21-00625-f003:**
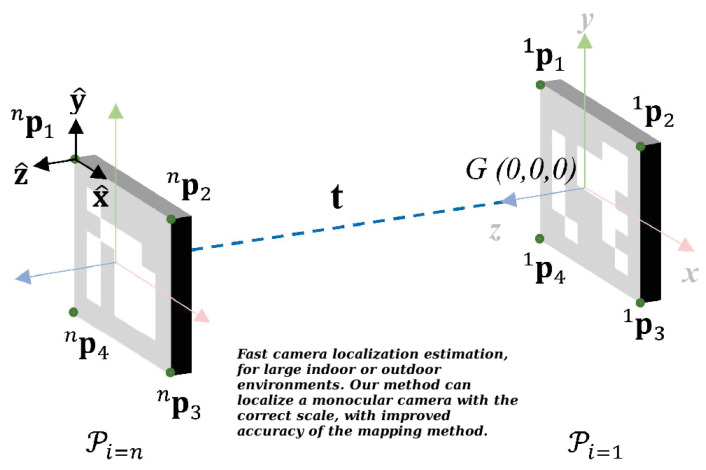
Process to compute $P_{i}$ for a marker set with *n* markers.

**Figure 4 sensors-21-00625-f004:**
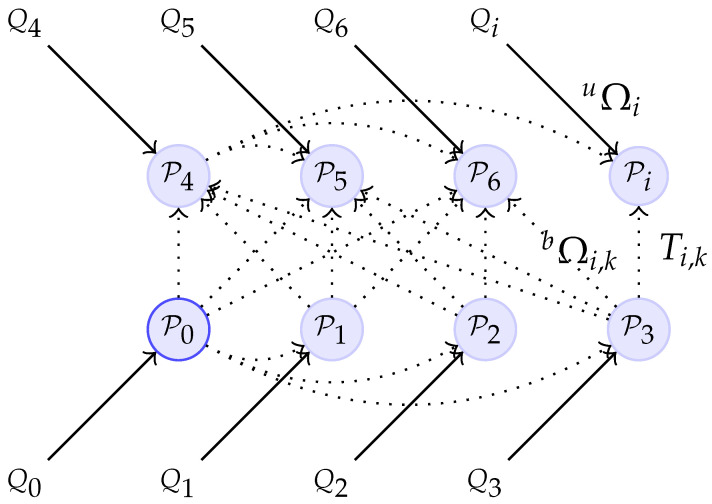
Pose graph for fusion problem. ${}^{u}\Omega_{i}$ and ${}^{b}\Omega_{i,k}$ are the information matrices for unary and binary edges respectively. $P_{0}$ represents the origin of the marker map, G(0,0,0).

**Figure 5 sensors-21-00625-f005:**
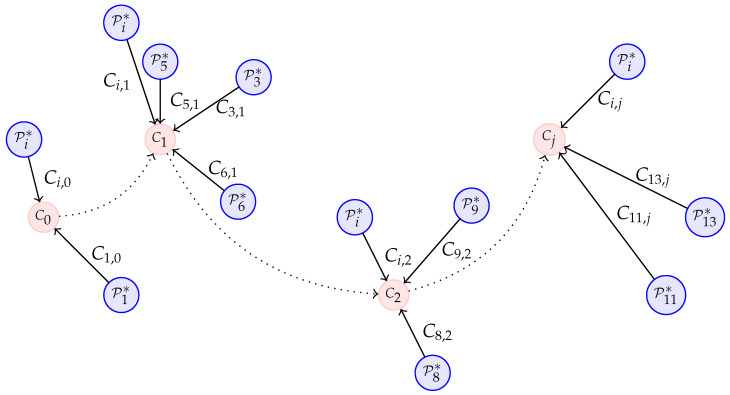
Poses graph for the camera localization problem.

**Figure 6 sensors-21-00625-f006:**
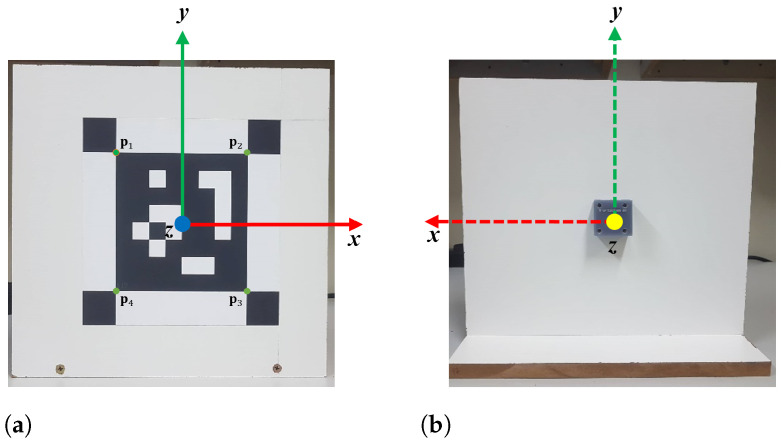
Smart marker: a square planar fiducial marker in front (**a**); and an embedded pose measurement system (PMS) unit with their coordinates systems at the back (**b**).

**Figure 7 sensors-21-00625-f007:**
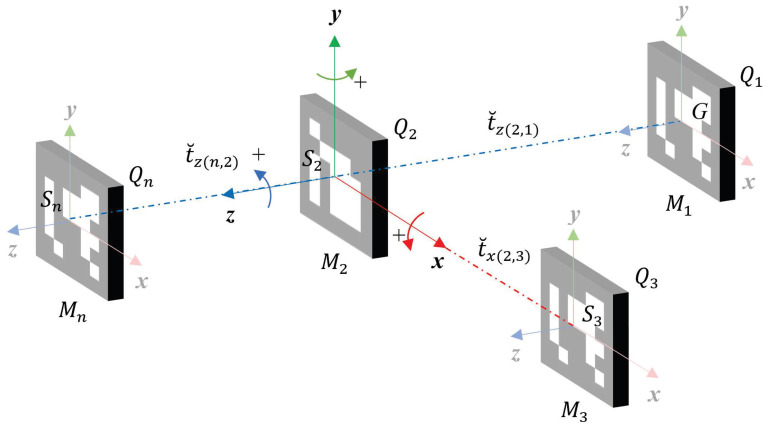
Possible configurations of markers within a scene; coordinate systems and measurements obtained with the PMS units.

**Figure 8 sensors-21-00625-f008:**
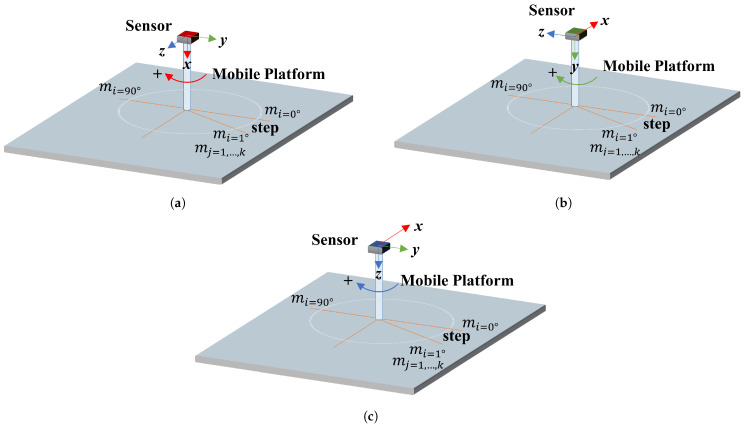
Schematic for evaluating the accuracy, repeatability, and total error, in (**a**) *x*–*roll*, (**b**) *y*–*pitch*, and (**c**) *z*–*yaw* axes of the orientation sensors.

**Figure 9 sensors-21-00625-f009:**
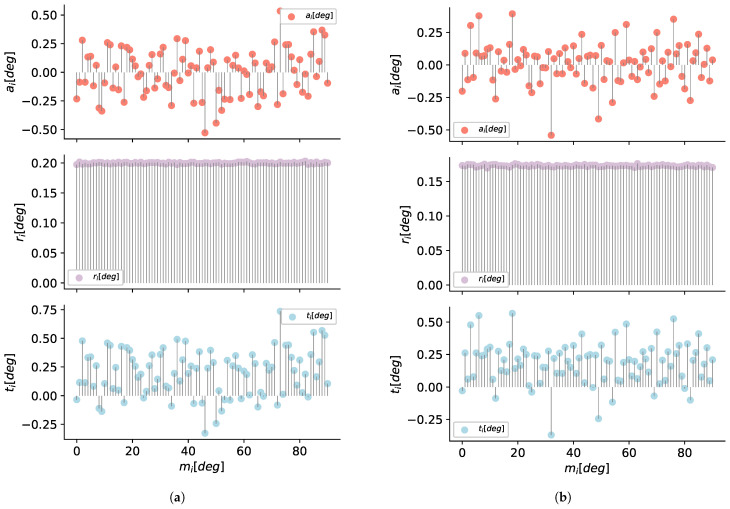
Results of accuracy, repeatability, and total error, for (**a**) *x*–roll, (**b**) *y*–pitch, and (**c**) *z*–yaw, axes of the LSM9DS1.

**Figure 10 sensors-21-00625-f010:**
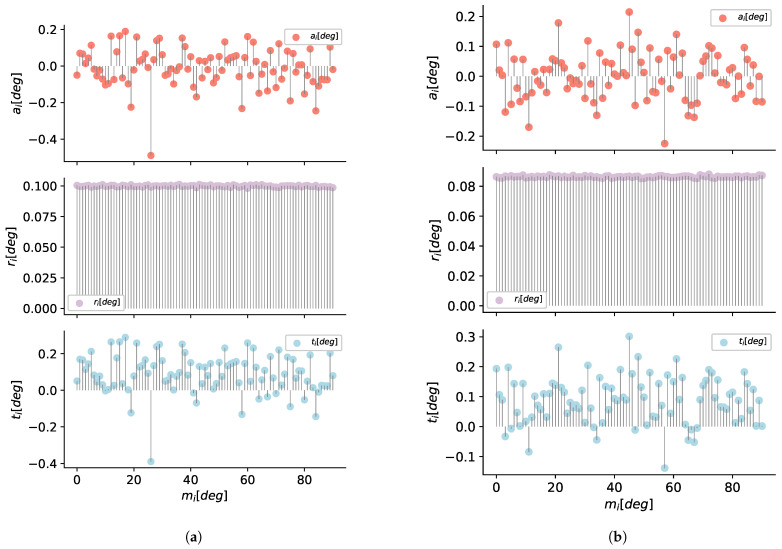
Results of accuracy, repeatability, and total error, for (**a**) *x*–roll, (**b**) *y*–pitch, and (**c**) *z*–yaw, axes of the PCEVDL16I.

**Figure 11 sensors-21-00625-f011:**
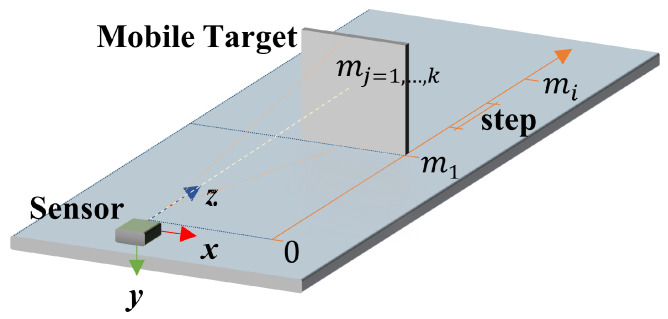
Schematic for evaluating accuracy, repeatability, and total error, in range of the axis of the position sensors.

**Figure 12 sensors-21-00625-f012:**
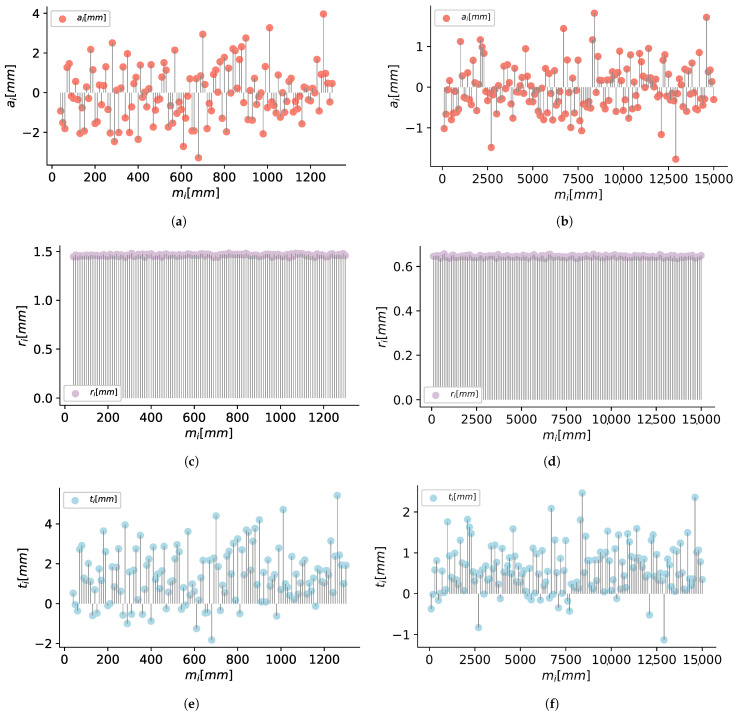
Results of (**a**) accuracy, (**c**) repeatability, and (**e**) total error for the sensor VL53L1X and (**b**,**d**,**f**) for the distance meter GLM80.

**Figure 13 sensors-21-00625-f013:**
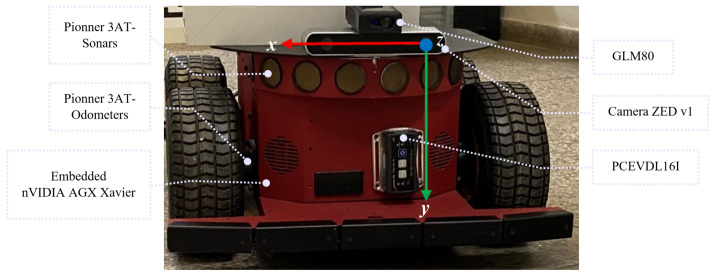
Hardware used for the experiments consisted of a Pioneer AT robot with the ZED stereo camera mounted on its top, and with the coordinate system origin referenced to the left camera center. The *x* axis is to the right of the robot (red arrow), *y* is pointing down (green arrow), and *z* is to the front (forward) of the robot (a right-hand coordinate system is used).

**Figure 14 sensors-21-00625-f014:**
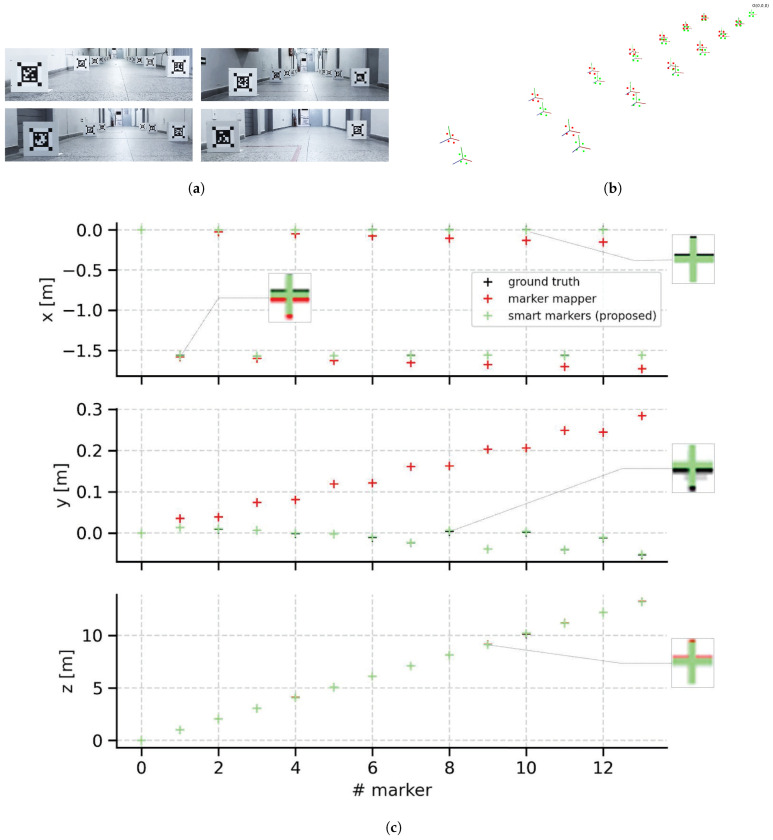
Evaluation of the mapping stage with smart markers with dataset **M-Seq1/hall**. Snapshots of smart markers distributed on the environments (**a**). Marker maps comparison (**b**). Three-dimensional reconstruction (**c**) in which red plus markers represent the Marker Mapper position estimates, green markers represent smart marker (our method) position estimates, and black are ground truth position estimates.

**Figure 15 sensors-21-00625-f015:**
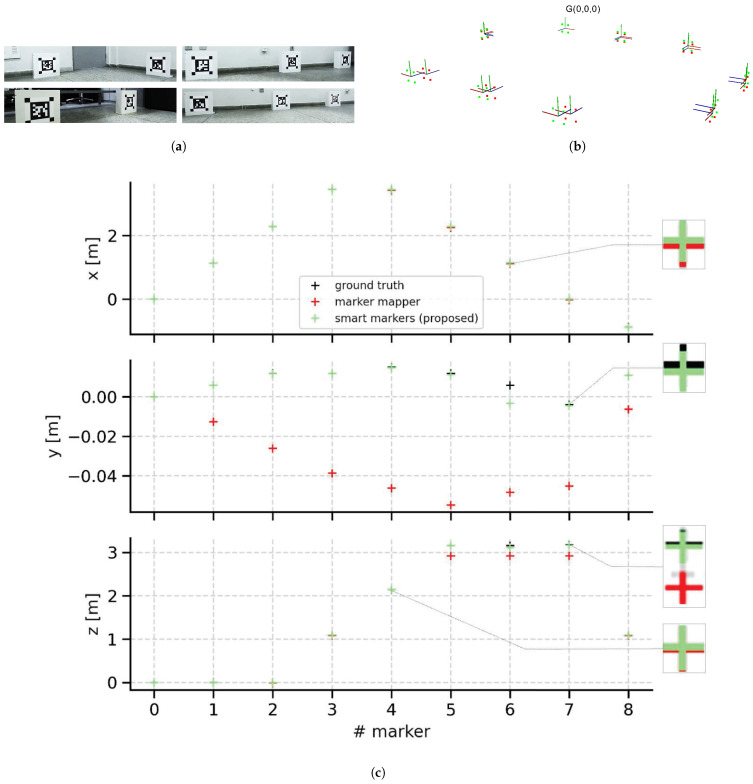
Evaluation of the mapping stage with smart markers with dataset **M-Seq1/hall**. Snapshots of smart markers distributed in the environments (**a**). Marker maps comparison (**b**). Three-dimensional reconstruction (**c**) in which red plus markers represent the Marker Mapper position estimates, green markers represent smart marker (our method) position estimates, and black are ground truth position estimates.

**Figure 16 sensors-21-00625-f016:**
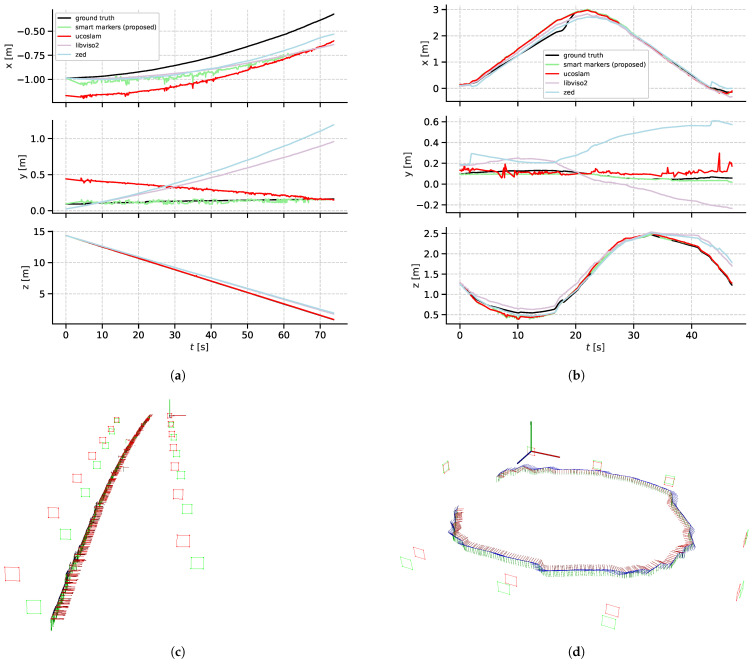
Camera localization evaluation, **T-Seq3/hall** dataset (**a**,**c**) and **T-Seq4/room** dataset (**b**,**d**). Comparison of camera trajectories (**a**,**b**) computed with UcoSLAM, LibViso2, ZED tracking API, and our method; and ground truth. In three dimensional views (**c**,**d**) we show our camera path, in which red squares represent the Marker Mapper markers and green ones are the poses of smart markers as computed by the first stage of our method.

**Table 1 sensors-21-00625-t001:** Results of the accuracy, repeatability, and total error for the sensors: LSM9DS1, PCEVDL16I, VL53L1X, and GLM80.

Sensor	Axis	Accuracy	Repeatability	Total Error
LSM9DS1	*x*–roll [0–90${}^{\circ }$]	±0.53	±0.20	±0.73
	*y*–pitch [0–90${}^{\circ }$]	±0.52	±0.17	±0.69
	*z*–yaw [0–90${}^{\circ }$]	±0.72	±0.37	±1.09
PCEVDL16I	*x*–roll [0–90${}^{\circ }$]	±0.2	±0.10	±0.3
	*y*–pitch [0–90${}^{\circ }$]	±0.21	±0.08	±0.27
	*z*–yaw [0–90${}^{\circ }$]	±0.50	±0.19	±0.69
VL53L1X	range [40–1300 mm]	±3.35	±1.48	±4.83
GLM80	range [100–15,000 mm]	±1.81	±0.65	±2.46

**Table 2 sensors-21-00625-t002:** Descriptions of video sequences for mapping and localization experiments.

Dataset Name	Duration [s]	f.p.s	Resolution [px]
M-Seq1/hall	135	15	2208 × 1242
M-Seq2/room	51	60	1280 × 720
T-Seq3/hall	50	15	2208 × 1242
T-Seq4/room	9	60	1280 × 720

**Table 3 sensors-21-00625-t003:** Accuracy of the generated map of markers. RPE (RMSE) computed from translational and rotational pose parts, and processing times for datasets **M-Seq1/hall** and **M-Seq2/room**.

Method	Dataset	${\mathrm{RPE}}_{\mathrm{trans}}$ [m]	${\mathrm{RPE}}_{\mathrm{rot}}$ [rad]	Comp. Time [ms]
Marker Mapper	M-Seq1/hall	0.045	0.031	**7420**
	M-Seq2/room	0.107	0.021	**890**
**smart markers (proposed)**	M-Seq1/hall	**0.006**	**0.03**	7440
	M-Seq2/room	**0.013**	**0.019**	930

**Table 4 sensors-21-00625-t004:** Accuracy of the camera localization. ATE (RMSE) metric and RPE (RMSE) computed from translational and rotational pose parts, and processing times for datasets **T-Seq3/hall** and **T-Seq4/room**.

Dataset	Method	ATE [m]	${\mathrm{RPE}}_{\mathrm{trans}}$ [m]	${\mathrm{RPE}}_{\mathrm{rot}}$ [rad]	Comp. Time [ms]
T-Seq3/hall	UcoSLAM	0.299	0.014	0.005	4.12
	LibViso2	0.685	0.010	0.0007	**2.33**
	ZED Camera API	0.791	0.008	0.0004	11.27
	**Smart Markers** **(proposed)**	**0.137**	**0.004**	**0.0001**	3.76
T-Seq4/room	UcoSLAM	0.206	0.035	0.018	3.72
	LibViso2	0.258	0.023	0.014	**2.68**
	ZED Camera API	0.393	0.027	**0.010**	10.23
	**Smart Markers** **(proposed)**	**0.124**	**0.013**	0.015	2.85

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
