# Peer review of "Smart Artificial Markers for Accurate Visual Mapping and Localization"

_sensors, 2021, doi:10.3390/s21020625_

Round 1
Reviewer 1 Report
In the paper, the authors propose to improve the accuracy of map construction and camera localization within the pre-built map using Smart Marker. Smart marker consisting of PMS unit and a binary visual pattern was designed as shown in Figure 4. The two experiments were carried out showing that the approach decreases the Relative Pose Error (RPE) in the mapping phase and absolute Trajectory Error (ATE) for the camera localization phase. In the Section 4.1.1, the authors should list the system measurement error of laser-ranging sensors. Could the relative distance between two consecutive markers be used directly to build marker map if the two ToF sensors output more accurate relative translations along x and z direction? In the situation, the fusion process in building the marker’s map will be more simply. As discussed in Section 7, there are some limitation in aligning Smart Markers, for example, the markers must be placed in the same (x, z) plane. The authors should discuss the data fusion in detail if (x, y) plane of Smart Markers are parallel and their y height values may be greatly different given that they can be also detected by the camera. In the paper, the marker gives more measurement data by IMU and TOF sensors. Then the data fusion algorithm is used to obtain an optimal pose for each marker in the map. As discussed above, if the TOF sensors output more accurate relative distance, only y height value should be computed, so the authors should discuss this in the Section 5.
Author Response
Please, see enclosed PDF.

Reviewer 2 Report
In this work, the authors proposed to improve the accuracy of map construction using artificial markers (mapping method) and camera localization within this map (localization method) by introducing a new type of artificial marker that they call Smart Marker. The mapping method estimates the markers’ poses from a set of calibrated images and orientation/distance measurements gathered from the PMS unit. The proposed localization method can localize a monocular camera with the correct scale, directly benefiting from the improved accuracy of the mapping method.
The recommendations are as follow:
- In subsection 4.1.2. PMS accuracy representation, the authors claim “ Our proposed PMS unit is made up of low-cost sensors”, the suggestion is to briefly describe in this subsection the model and type of these sensor.
- In section 5.1 Please provide a picture or photograph for the system mounted on top of a Pioneer 3AT robot.
Author Response
Please, see enclosed PDF.

Reviewer 3 Report
This paper approaches the problem of VSLAM using coded targets with additional sensors which provide targets pose. Based on this set of markers they compute camera locations and attitude.
There is some novelty in the use of this concept of smart target, although, some concern arises on the accuracy of the resulting poses with low cost sensors.
English style is suitable but some parts needs revision. There are some parts which are not entirely clear.
There is a lack of details in some parts, mainly in the camera location process.
In the experiments section they compared several techniques. For the camera location, however, it is not clear whether the techniques are comparable, due the type of information used in each one. For instance, the proposed technique uses several markers with known poses, thus providing very rigid information on the global reference system. It is not clear how this information was used by the other techniques.
The discussion of the results is also limited; some findings, such as the discrepancies in y axis in some techniques, should be discussed and explained.
I´ve attached a pdf file with several comments, suggestions and notes, that should be treated by the authors in case of resubmission.

Author Response
Please, see enclosed PDF.

Round 2
Reviewer 1 Report
The manuscript has been significantly improved and can be accepted.
Author Response
Please, see attached file.

Reviewer 3 Report
The manuscript has been significantly improved and can be accepted.
I have minor comments that can be approached by the authors and checked by the editor.
All comments were approached in some extent.
Some minor issues to address:
Row 70 “…. avoid the significant errors originating from erroneous tracking of keypoints ….”. Indeed, the wrong matches are excluded in the estimation process, thus the final results are usually achieved without erroneous keypoints. Also, because many keypoints are detected the effect of wrong matches are almost negligible.
Row 162. “… f the considerable errors (i.e. ≈ 5 meters) inherent to this technology …”
Please, revise this statement. This level of errors is achieved only in ordinary navigation grade GPS. Centimetre (and even millimetre) accuracies can be achieved with Real Time Kinematics (RTK) or with other post-processed techniques, for instance DGPS or PPP. You concern with lack of signals in indoor applications is correct, but the statement on accuracy must be revised.
Some sentences recently inserted need some polish: for instance, in row 561.
The paper is now suitable for publication.
Author Response
Please, see attached file.
